# Body Image, Anxiety, and Bulimic Behavior during Confinement Due to COVID-19 in Mexico

**DOI:** 10.3390/healthcare9111435

**Published:** 2021-10-25

**Authors:** Gisela Pineda-García, Aracely Serrano-Medina, Estefanía Ochoa-Ruíz, Ana Laura Martínez

**Affiliations:** Departamento de Posgrado, Facultad de Medicina y Psicología, Universidad Autónoma de Baja California, Tijuana 22300, Mexico; gispineda@uabc.edu.mx (G.P.-G.); estefania.ochoa@uabc.edu.mx (E.O.-R.); ana.laura.martinez.martinez@uabc.edu.mx (A.L.M.)

**Keywords:** bulimic behavior, anxiety, body image, COVID-19, home confinement, body dissatisfaction

## Abstract

Background: The potential impact of the COVID-19 pandemic on weight, shape-related appearance behaviors (body image dissatisfaction), and bulimic symptoms in nonclinical participants is poorly evaluated. This study aimed to identify the relationship between labor status, confinement degree due to COVID-19, dissatisfaction with body image, and anxiety and to discover its effect on bulimic behavior in Mexican adults. Methods: A cross-sectional study was conducted with a non-probabilistic sample of 276 participants via an online survey. Questions addressed their anxiety, body image dissatisfaction, and bulimic behavior. Results: The path analysis showed direct effects on the confinement degree (*β* = −0.157); of the labor situation (*β* = −0.147) and gender (*β* = 0.129) on anxiety; of dissatisfaction on bulimic behavior (*β* = 0.443) and anxiety about bulimic behavior (*β* = 0.184); and dissatisfaction (*β* = 0.085). Conclusions: The confinement, gender, and labor status are predictors of anxiety, while anxiety and body dissatisfaction directly influence bulimic behavior.

## 1. Introduction

A novel severe acute respiratory syndrome (SARS)-like coronavirus (SARS-CoV-2) recently emerged in 2020 and rapidly spread in humans causing the COVID-19 pandemic, [1]. The vertiginous spread of the virus has affected the population in different aspects, ranging from the economy to behavior and unfortunate psychosocial implications [2]. No specific therapeutics are available, and current management includes travel restrictions, home confinement or social isolation, and supportive medical care [3]. The lockdown measures have had a great impact on everyday life, often associated with a negative influence on psychological wellbeing. These circumstances have exacerbated a series of psychological and psychopathological conditions, including emotional exhaustion, irritability, anxiety, increased anger, depressive symptoms, and post-traumatic stress disorder [4]. Body image refers broadly to individuals’ subjective experiences with regard to their appearance and includes various perceptual, cognitive-evaluative, and affective aspects, which in turn influence behavioral and psychosocial functioning [5]. Disturbances in body image are part of the diagnostic criteria for anorexia nervosa [6] and bulimia nervosa and are thought to play a critical role in the maintenance of the psychopathology of other eating disorders (ED), such as extreme dietary restriction, binge eating, purging, low weight, and associated eating-related concerns [5]. Anxiety may be one of the undesirable consequences of confinement; in a meta-analysis conducted with studies that examined anxiety among the general population during the COVID-19 pandemic, several studies were evaluated with a total sample size of 63,439 participants, revealing a 39% prevalence of anxiety [7]. Bulimic behavior is highly related to anxiety, whereby female patients that suffer from bulimia nervosa have higher scores of anxiety and greater difficulties with emotional regulation than healthy female subjects controls [8]. Individuals with ED have been documented with increased engagement in dietary restriction, exercise, binge eating, and purging relative to before the pandemic [9]. On the other hand, there is a report in which it was observed that ED prevalence increased during the COVID-19 pandemic, strongly associated with anxiety and intolerance of uncertainty [10]. However, the potential impact of the COVID-19 pandemic on shape-related appearance behaviors (body image dissatisfaction), and bulimic symptoms in nonclinical participants are poorly evaluated. In this study, a model is proposed where the interaction between the variables confinement degree and labor conditions predicts anxiety level (interaction no reported in other studies). Moreover, anxiety and body dissatisfaction explain characteristics of bulimic behavior in a Latin American context, specifically in samples of the Mexican general population. Thus, the current research aimed to describe the levels of anxiety, body image dissatisfaction, and bulimic behavior in Mexican adults; identify if there is a relationship between working status, degree of confinement due to COVID-19, dissatisfaction with body image, and anxiety; and know the effect of these on bulimic behavior. In accordance with previous studies, the following hypotheses were formulated: (a) that the degree of confinement and labor status has a direct effect on anxiety; (b) that anxiety and corporal dissatisfaction has a direct effect on bulimic behavior.

## 2. Methods

### 2.1. Study Design and Participants

Cross-sectional quantitative data were collected from participants via an online survey between 1 and 31 May 2020, the final month of COVID-19 restrictions in Mexico before the reopening of activities across all states. Members of the general public residing in Mexico who were over 18 years of age were invited to participate, whereas people who were not of Mexican nationality and who reported using drugs were excluded. The sample was a non-probabilistic, of the type “snowball,” where the respondents were recruited from social media advertisement. The G*Power 3.1.9.7 [11] software was used to determine the statistical potency and the size of the effect, a value of 1.00 was obtained for the first and a value of f^2^ = 0.33 for the second (medium, near to a large effect >0.35), thus having an R^2^ = 0.248, α = 0.05 The research managers sent the invitation to respond to the survey to students, relatives, and acquaintances. Informed consent was obtained from the volunteers participating in the study before starting the online survey. The online survey included a battery of measures and took 15 min to complete. Participants completed demographic questions providing their age, gender, employment, and current living situation. All procedures performed in this study were in accordance with the institutional review board (approval number D249) and with the World Health Organization (WHO) code of ethics (Declaration of Helsinki). The ethical recommendations for non-invasive procedures in the research were followed according to the Mexican Society of Psychology.

### 2.2. Labor Status and Degree of Confinement

The labor status was asked with the question: Do you currently have a job? (a. no; b. yes). The degree of confinement was measured with the question What is your degree of confinement? (a) I practically do not leave home (qualified answer with the value of 5); (b) I only leave my house to do the most indispensable (assigned value of 4); (c) Most of the time I am at home, but I go out to visit relatives or leave for other reasons (for example: walk the pet), (assigned value of 3); (d) My work does not allow me to stop leaving, but the rest of the time I remain in confinement, (assigned value of 2); and (e) I continue with my activities as usual (assigned value of 1). Response options were graded in such a way that a higher value implies a higher degree of confinement.

### 2.3. Anxiety Assessments

Anxiety symptoms were measured using the Goldberg Anxiety Scale’s nine-item version [12], a well-established screening instrument with nine dichotomous items (yes and no) to identify psychiatric anxiety disorders in clinical and nonclinical samples. It presents sufficient psychometric properties with a cutoff point ≥4 for the identification. The scale was adapted to Spanish with adequate psychometric properties [13]; in the sample participants, it has Cronbach’s Alpha of α = 0.697 and two factors that explain 44% of the variance.

### 2.4. Body Image Dissatisfaction

Perceived and ideal figures were determined using the group of silhouettes proposed by Thompson and Gray, 1995 [14], covering a continuum of weight from an undernourished or emaciated figure to one with obesity, passing through a normal weight silhouette and presented evidence of concurrent validity equal to 0.71 with body weight and test–retest reliability *r* = 0.78. In samples of Latin American participants, concurrent validity with BMI indicates values *r* = 0.81 in women and *r* = 0.78 in men [15]. The silhouettes group was shown to participants at two different moments to evaluate the perception of their current body type and their desired body type. The level of dissatisfaction or satisfaction with their body image was measured as a function of the difference between values given for their perceived figure and their ideal figure. A difference equal to zero indicated satisfaction, a positive difference represented dissatisfaction due to a desire to be thinner, and a negative difference indicated dissatisfaction due to a desire to be thicker. A bigger difference denoted a greater degree of dissatisfaction [16].

### 2.5. Bulimic Behavior

Bulimic behavior was measured using the bulimia nervosa and preoccupation with food subscale (Cronbach’s alpha of α = 0.80) of the Eating Attitudes Test-26 (EAT-26) [17], which is a screening instrument for detection of the risk of ED, the scale was adapted to Spanish by Castro, 1991 [18]. It presented a general Cronbach’s alpha coefficient of α = 0.77, with a solution of four factors that explained 43% of the variance in the sample participants. The scale has 26 Likert-type items (never, rarely, sometimes, often, very often, and always); the first three responses are scored as 0, the fourth response is scored as 1, the fifth response is scored as 2, and the sixth response is scored as 3. To measure bulimic behavior, a summation was carried out with six items: (3) food is a common concern for me; (4) I have suffered binge attacks in which I had the feeling of not being able to stop eating; (9) I vomit after eating; (18) I have the impression that my life revolves around food; (21) I spend too much time thinking about food; and (25) I enjoy trying new and tasty foods. Scores were summed to produce a total EAT-26 score (range = 0–18). A score equal to zero was considered without risk, while scores ranging from 1–6 indicated low bulimic behavior, scores ranging from 7 to 12 indicated medium bulimic behavior, and scores ranging from 13 to 18 indicated high bulimic behavior.

### 2.6. Statistical Analysis

In the data analysis, descriptive (mean and standard deviation) and inferential statistics were used, comparisons with percentage distributions using the SPSS software (IBM. Armonk, NY, USA) for Windows version 22. Path analysis was designed with support from the AMOS module for SPSS. For the path analysis, the dichotomous variables gender and employment status were converted into a dummy (woman = 1, man = 0; works = 1, does not work = 0). The anxiety and bulimic behavior variables were described in categorical form, whereas a continuous form was used for path analysis. The maximum-likelihood method was used for estimating model parameters. The values of Chi-square (*Χ*^2^), Normed Fit Index (NFI), Tucker–Lewis Index (TLI), Root Mean Square Error (RMR) and Root-Mean-Square Error of Approximation (RMSEA), were reported as model fit indicators.

## 3. Results

### 3.1. Sample Characteristics

A total of 281 Mexican individuals answered the online survey; the participants’ sample characteristics included an average age of 27.8 years (SD = 11.1), with 70.0% being female participants and 30% being male participants. Moreover, 47% of the participants reported having a paid job, 55% of the participants were students, 4% reported studying and working, and the remaining 41% were people dedicated to various trades and professions. The data associated with the sample description can be found in Table 1.

### 3.2. Anxiety Levels, Body Image Dissatisfaction, and Bulimic Behavior

To address the first aim, the levels of anxiety, body image dissatisfaction, and bulimic behavior are described in Table 1. It can be observed that almost 31% of the total sample presented anxiety, with the number of female participants being 8% higher than that of male participants. In Figure 1, discrepancies can be observed in the levels of body image dissatisfaction (17% in male participants and 10% in female participants), as well as in the degrees of negative or positive dissatisfaction, with more participants being dissatisfied due to a desire to be thinner with respect to a desire to be thicker (negative dissatisfaction). The number of respondents indicating the former was higher among female participants (77%, adding all degrees of positive dissatisfaction) compared to male participants (66%), who did not present cases with scores of 5 or 6. In the case of dissatisfaction due to a desire to be thicker, there were also differences observed between male participants (17%) and female participants (12%), the analysis of the means, also in Table 1, indicates significant differences by gender (*t*(279) = −3.566, *p* = 0.000). Regarding bulimic behavior (Table 1), all participants presented a risk of exhibiting bulimic behavior: 88% at a low level, 12% at a medium level (12% in female participants and 11% in male participants), and 0.5% at a high level (where only female participants manifested a high level of bulimic behavior).

### 3.3. Effect of Anxiety, Body Image Dissatisfaction, Confinement Degree, and Labor Status on Bulimic Behavior

To identify if there was a relationship among labor status, confinement, anxiety, and body image dissatisfaction with respect to bulimic behavior, a partial correlation analysis was applied to the variables, controlling for age and gender (Table 2). Based on these findings and theoretical assumptions, a path analysis was carried out to determine the effect sizes of the predictor variables (degree of confinement, labor situation, and gender) on anxiety and body dissatisfaction, and of these on the main exogenous variable (bulimic behavior). In Figure 2, a model with adequate goodness-of-fit values (X^2^ = 13.697, *p* = 0.052, NFI = 0.898, TLI = 0.877, RMR = 0.085 and RMSEA = 0.060) is shown, where the following can be seen (if our assumptions are correct and not forgetting the cross-cutting nature of the data): A direct effect of gender (*β* = 0.129, 95% Confidence Interval CI: −0.065 to 0.680), labor status (*β* = −0.147, 95% CI: −0.779 to −0.081) and degree of confinement (*β* = −0.157, 95% CI: −0.647 to −0.132) on anxiety, whereby, being a woman increases the presence of anxiety, having a job led to a decrease in anxious state decreases, while a decrease in the degree of confinement led to a decrease in an anxious state. A direct effect of gender (*β* = 0.199, 95% CI: 0.305 to 1.114) and anxiety (*β* = 0.085, 95% CI: −0.049 to 0.203) on body image dissatisfaction, so being a woman and anxiety increases dissatisfaction. A direct effect of body image dissatisfaction (*β* = 0.443, 95% CI: 0.873 to 1.395) and anxiety (*β* = 0.184, 95% CI: 0.079 to 0.648) on bulimic behavior. Whereby an increase in body dissatisfaction and anxiety led to increased bulimic behavior. The explained variance for the main dependent variable was 24.8%, this percentage explains the variability in the bulimic behavior with the linear relation between the degree of anxiety and body dissatisfaction, expressed in the following formula:R^2^ = p_31_ × r_31_ + p_32_ × r_32_ = 0.181 × 0.230 + 0.444 × 0.464 = 0.247646(1)
where p_31_ represents the value of the path coefficient of the anxiety on bulimic behavior, r_31_ the correlation between these variables; p_32_ represents the value of the path co-efficient of body dissatisfaction on bulimic behavior and r_32_ the correlation between these variables

The findings confirmed the research hypothesis.

## 4. Discussion

In periods of uncertainty, as experienced during confinement due to the COVID-19 pandemic, people are more vulnerable to suffering from mental illness and presenting comorbidity with other diseases [19]. Typically, women have more significant anxiety than men [20], as in this study where the model identifies an effect of sex on anxiety: being female increases this condition by 0.129. The role of anxiety in explaining body dissatisfaction has been relevant in the literature as in the Fitzsimmons-Craft and Bardone-Cone study who point out that anxiety associated with body surveillance is a factor predicting dissatisfaction [21], the same predictive finding was found in this research; however, the body’s surveillance variable was not measured, so further research is needed. It is striking that all the participants showed bulimic behavior, albeit with most exhibiting a low level and only 12% of participants exhibiting a medium level. This finding is important for participants’ food health since certain characteristics of bulimic behavior, such as binge eating, display comorbidity with obesity [22] and anxiety [16]. At the beginning of the century, body image dissatisfaction was identified as a normative characteristic of women in the United States [23], presenting as a risk factor for disorderly eating behavior [24]. In Mexico, the same normative characteristic seems to be present, with the present study’s results describing high levels of body image dissatisfaction in participants: many of them (particularly female participants) expressed the desire for a slimmer silhouette. The results confirm previous research trends where high percentages of body image dissatisfaction were reported in both Mexican men and women, with the former desiring a slim muscular figure and the latter desiring slim figures defined with breasts and glutes [25,26], thus the significant differences found in this variable confirm once again the discrepancies by gender described in previous international research [27]. Body image dissatisfaction has been investigated in association with bulimic behavior [23]; however, no reports have been published so far on the Mexican population during the period of COVID-19. In a Lebanese population, the interest in and concern for body image were studied, finding no effects on eating behavior during the quarantine period [28], in contrast to the findings of our study reporting that an increase in the degree of body image dissatisfaction led to an increase in bulimic behavior (0.463) in the study sample. However, these findings agree with cross-sectional studies on nonclinical samples of American college women, where the same relationship was shown between the variables before COVID-19 [29]. Among the main contributions of this research was identifying the effect of the degree of confinement and labor status on the participants’ anxiety, with the subsequent direct effect on bulimic behavior, mediating the relationship among confinement, work situation, and bulimic behavior. Among the participants, having a job reduced anxiety, leading to the hypothesis that having job stability (ensuring economic income) decreases anxious states. However, a lower degree of confinement increased anxiety, potentially explained through fear of contagion by no maintaining social distancing and following stay-at-home guidelines proposed by the Mexican Ministry of Health [30]. In this sense, it is appropriate to mention that there is no unemployment insurance in Mexico, unlike in the United States and some European countries. At the same time, many companies do not allow paid work rest, in addition to 28 million Mexicans having informal jobs [31]. The relationship between anxiety and the risk of suffering from bulimia was confirmed in the present study, establishing a direct effect, suggesting that overeating is a type of negative behavior for coping with uncomfortable emotions, which some authors call emotional eating [32]. These findings underscore the importance of the design, implementation, and evaluation of interventions promoting mental and nutritional health through strategies such as mindfulness, leading to a health process action model that can be combined with traditional and alternative pharmaceutical prescriptions. A limitation of the present study is its single-measure cross-sectional design, which made it impossible to compare anxiety, body dissatisfaction, and bulimic behavior within a context outside the period of confinement due to the COVID-19 pandemic. In contrast, a longitudinal design would allow for a better assessment of the relationship among variables. The small sample size, especially from male participants, and the sample’s non-probabilistic nature made it impossible to generalize the broader population results. Furthermore, the sample was not representative of the population from which it was extracted, as respondents of the electronic survey were limited to people with a mobile device or computer and access to the internet. Lastly, residual confounding bias was also present, as there could be other factors related to anxiety and bulimic behavior that were not evaluated in the present study.

## 5. Conclusions

According to our results, if our assumptions are correct and not forgetting the impossibility of generalizing the results to the population, we conclude that the degree of confinement, gender, and labor status are predictors of anxiety. In contrast, anxiety and body image dissatisfaction directly influence bulimic behavior, manifested in all study’s participants during the period of confinement due to COVID-19.

## Figures and Tables

**Figure 1 healthcare-09-01435-f001:**
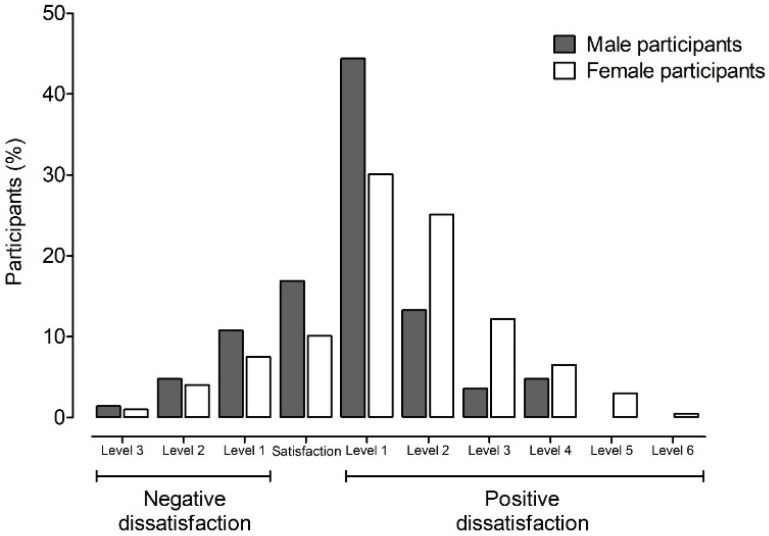
Body image dissatisfaction in Mexican adults during confinement due to COVID-19. Each bar represents the level of negative and positive dissatisfaction in male and female participants, measured as the difference between perceived and ideal figures.

**Figure 2 healthcare-09-01435-f002:**
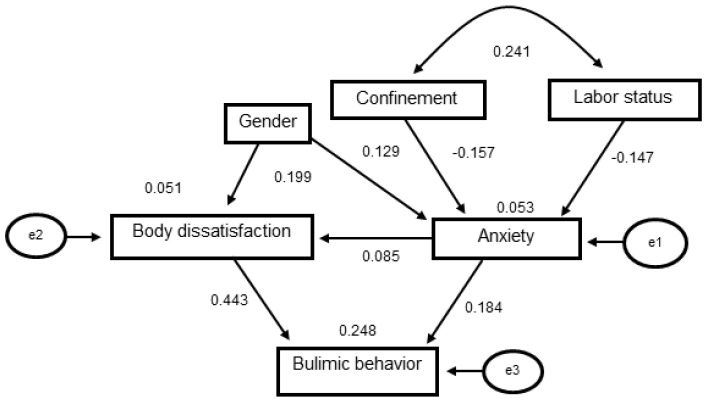
The bulimic behavior model explained from anxiety and body dissatisfaction. Body dissatisfaction is explained from anxiety and gender. Anxiety is explained from confinement and labor status. The rectangles represent the observed variables; the ovals represent the errors associated with the endogenous variables: e1 is the measurement error for anxiety; e2 for body dissatisfaction and e3 for the bulimic behavior.

**Table 1 healthcare-09-01435-t001:** Sample characteristics grouped by gender. Variable distributions are reported as a *n* (%) unless otherwise specified.

Sample Characteristic	Women *n* = 194	Men *n* = 82	Total *n* = 276
Demographics			
Age, mean (SD)	29.15 (11.37)	24.55 (10.04)	27.8 (11.16)
Minimum	18	18	18
Maximum	62	66	66
18–40	160 (82.5%)	76 (92.7%)	236 (85.5%)
41–66	34 (17.5%)	6 (7.3%)	40 (14.5%)
Labor status			
Employed	89 (45.9%)	42 (51.2%)	131 (47.5%)
Unemployed	105 (54.1%)	40 (48.8%)	145 (52.5%)
Degree of confinement			
Continues with activities as usual	1 (0.5%)	3 (3.6%)	4 (1.4%)
Work does not allow to stop going out, but the rest of the time stay in confinement	14 (7.2%)	10 (12.2%)	24 (8.7%)
Most of the time stay at home, but go out to visit relatives or go out for the other reasons	16 (8.2%)	4 (4.9%)	20 (7.2%)
Only leave the house to do the most essential	163 (84.0%)	65 (79.3)	200 (82.6%)
Anxiety			
Mean (SD)	27.00 (5.61)	25.56 (5.9)	26.57 (5.73)
Minimum	13	13	13
Maximum	39	39	39
Presence	64 (33.0%)	21 (25.6%)	85 (30.8%)
Absence	130 (67.0%)	61 (74.4%)	191 (69.2%)
Body image			
Dissatisfaction, mean (SD)	1.47 (1.65)	0.73 (1.39)	1.25 (1.61)
Minimum	−3	−3	−3
Maximum	6	4	6
Satisfaction	20 (10.3%)	14 (17.1%)	34 (12.3%)
Desire to be slimmer	149 (76.8%)	54 (65.9.2%)	203 (73.6%)
Desire to be thicker	25 (12.9%)	14 (17.1%)	39 (14.1%)
Bulimic behavior			
Mean (SD)	4.38 (4.04)	3.70 (4.03)	4.17 (4.06)
Minimum	1	1	1
Maximum	24	18	24
Low bulimic behavior	169 (87.1%)	73 (89%)	242 (87.8%)
Medium bulimic behavior	24 (12.4%)	9 (11%)	33 (12.0%)
High bulimic behavior	1 (0.5%)	-	1 (0.4%)

SD: Standard Deviation.

**Table 2 healthcare-09-01435-t002:** Bivariate partial Pearson correlations among labor status, degree of confinement, anxiety, body dissatisfaction, and bulimic behavior, controlled for gender and age. A *p*-value < 0.05 was considered statistically significant.

Control Variables	Laboral Status	Degree of Confinement	Anxiety	Body Image Dissatisfaction	Bulimic Behavior
Laboral status					
Correlation	1.00				
Significance (2 tailed)					
Df	0				
Degree of confinement					
Correlation	−0.248 **	1.00			
Significance (2 tailed)	0.000				
Df	276	0			
Anxiety					
Correlation	−0.064	−0.146 *	1.00		
Significance (2 tailed)	0.287	0.015			
Df	276	276	0		
Body image dissatisfaction					
Correlation	0.123 *	−0.100	0.098	1.00	
Significance (2 tailed)	0.040	0.097	0.104		
Df	276	276	276	0	
Bulimic behavior					
Correlation	0.068	−0.131 *	0.228 **	0.459 **	1.00
Significance (2 tailed)	0.255	0.029	0.000	0.000	
Df	276	276	276	276	0

* Significant correlations at the 0.05 level. ** Significant correlations at the 0.001 level. Df: degrees of freedom.

## Data Availability

The data associated with the paper are not publicly available, but are available from the corresponding author on request.

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
