# Peer review of "Body Image, Anxiety, and Bulimic Behavior during Confinement Due to COVID-19 in Mexico"

_healthcare, 2021, doi:10.3390/healthcare9111435_

Round 1
Reviewer 1 Report
This study analyzed the associations among body Image, Anxiety and bulimic behavior in Mexico during the pandemic of COVID-19, and these association was also supported based on the data of residences. There are also some comments, which may be helpful for revising this manuscript.
- As this is a cross-sectional study, we cannot get any causal relationships. The title may be needed to be revised.
- As this study was conducted during the pandemic of COVID-19, some information about the different relationships during pandemic of COVID-19 or not was necessary.
- Similar to the last question, as I know, there are many studies explored these relationships, and I want to know this study adds what information to our knowledge.
- In this study, the authors analyzed 281 participants. I am not sure about if the sample size is enough to do this study. Some calculations about the sample size was needed in this study.
- The gender differences between body image were found in many previous studies, and it may be also necessary for the authors to analyze the associations by gender.
Reviewer 2 Report
The manuscript entitled "Effect of Body Image and Anxiety on Bulimic Behavior during Confinement due to COVID-19 in Mexico" is an interesting work due to a set of variables such as gender, body dissatisfaction, work status, individual isolation level and anxiety as predictors of behavior bulimics during the COVID-19 pandemic. The manuscript is generally well written. However, some suggestions may improved this work.
- Please add a rating scale to the measurement section 2.2, describing confinement. The best way to do this is to add the appropriate numbers that correspond to each sentence as an answer to the question of the degree of confinement.
- Please add the Cronbach's alpha coefficient to show reliability of each instrument used in the present study sample.
- Please, add range of scores for each variable in Table 1.
- Please, add the correlation analysis before SEM. A table with Pearson's r correlations (and p-value) between all continuous variables could be very helpful.
- Table 2 is unclear. First of all, it is suggested to describe the results as a text (do not use a table for such small data set). Between which variables the squared multiple correlation is presented? If bulimic behavior is dependent variable, what "correlation" 0.248 represents? How the 25% of variance explained was assessed? Please, add an equation and some reference to it. This issue should be fully explained.
- The GFI and AGFI are not recommended to use in SEM analysis. Could you remove it, and include such fit indices as TLI, NFI and SRMS?
Round 2
Reviewer 1 Report
I think the authors have explained my comments in a reasonable way, and I suggest to accept this manuscript in this version.